# Rapid Resolution of Life-Threatening Hypertriglyceridemia after Evinacumab Administration in a Pediatric HSCT Recipient: A Case Report

**DOI:** 10.3390/ph16081069

**Published:** 2023-07-27

**Authors:** Alice Fachin, Chiara De Carlo, Alessandra Maestro, Davide Zanon, Egidio Barbi, Natalia Maximova

**Affiliations:** 1Department of Medicine, Surgery and Health Sciences, University of Trieste, 34127 Trieste, Italy; alice.fachin@burlo.trieste.it (A.F.); egidio.barbi@burlo.trieste.it (E.B.); 2Department of Medicine, Surgery and Health Sciences, University of Udine, 33100 Udine, Italy; chiara.decarlo@asfo.sanita.fvg.it; 3Institute for Maternal and Child Health—IRCCS Burlo Garofolo, 34137 Trieste, Italy; alessandra.maestro@burlo.trieste.it (A.M.); davide.zanon@burlo.trieste.it (D.Z.)

**Keywords:** hypertriglyceridemia, dyslipidemia, evinacumab, ANGPTL3-inhibitors, lipid-lowering drugs

## Abstract

Evinacumab, a human monoclonal antibody against angiopoietin-like protein 3 (ANGPTL3), has recently been approved by the U.S. Food and Drug Administration as an add-on therapy for homozygous familial hypercholesterolemia (HoFH) in patients of 12 years and older. Its role as a triglyceride-lowering drug is also emerging in the literature. However, it has not been approved for this indication yet, neither in the adult nor in the pediatric population. We describe the case of a 10-year-old boy who underwent an allogeneic hematopoietic stem cell transplant for acute lymphoblastic leukemia complicated by chronic graft-versus-host disease (GVHD) and presented life-threatening refractory hypertriglyceridemia due to the concomitant use of ruxolitinib and sirolimus. After the failure of the insulin treatment and due to the technical impossibility of performing lipid apheresis, the child underwent evinacumab treatment, obtaining a dramatic rapid reduction in triglyceride and cholesterol levels. This is the first report of a pediatric patient younger than 12 years in Europe receiving evinacumab to treat severe hypertriglyceridemia. The therapy with angiopoietin-like proteins inhibitors has been effective, safe, and well-tolerated in our patient, suggesting that evinacumab may be used in the pediatric population when other therapeutic strategies are ineffective or contraindicated.

## 1. Introduction

Chronic graft-versus-host disease (GVHD) is one of the major complications of allogeneic hematopoietic stem cell transplantation (HSCT). It occurs in up to 70% of patients who have undergone allogeneic HSCT, representing a leading cause of non-relapse morbidity and mortality. While the pathophysiological mechanisms have been better understood in acute GVHD, in chronic GVHD, there is still little knowledge. We know that chronic GVHD resembles an autoimmune syndrome, being characterized by an immune dysregulation involving both T-cell and B-cell immunity and a defective tolerance mechanism [1]. Due to their lymphopenic and anti-inflammatory properties, historically, systemic corticosteroids have been the standard first-line treatment for chronic GVHD; particularly, a starting dose of 1 mg/kg/die of prednisone (or an equivalent dose of methylprednisolone) has typically been used in this setting. Systemic glucocorticoid therapy could be given alone or in association with other immunosuppressive agents, such as calcineurin inhibitors (e.g., tacrolimus), which typically prevent IL-2 production by T-cells. However, approximately 50% of patients may develop a corticosteroid refractory chronic GVHD, requiring second-line systemic treatments, with a response rate varying from 25 to 50%, with no single therapy being better than the other [2]. Choosing a specific drug as a second-line treatment in chronic GVHD generally depends on the patients’ comorbidities, specific organ involvement in the graft, and clinicians’ experience. Particularly, mTOR inhibitors (e.g., sirolimus), Janus kinase (JAK) inhibitors (e.g., ruxolitinib or baricitinib), Bruton’s tyrosine kinase inhibitors (e.g., ibrutinib), proteasome inhibitors (e.g., bortezomib), and anti-CD20 monoclonal antibodies (e.g., rituximab) have shown promising efficacy with an adequate safety profile [3]. Among the adverse effects of these drugs, metabolic syndrome has frequently been documented in the literature [4]. Particularly, severe hypertriglyceridemia in patients treated with ruxolitinib (JAK1/2 inhibitor) in combination with sirolimus (mTOR inhibitor) has been reported both in adult and in pediatric populations, requiring a dose adjustment or drug withdrawal alone or in association with insulin drip or therapeutic plasma exchange [5,6]. In adult patients with severe hypertriglyceridemia who require a pharmacologic intervention, fibrates represent the first-line treatment. Omega-3 fatty acids, niacin, and statins in high doses have shown modest effectiveness [7]. Recently, novel lipid-lowering medications addressing new targets have been tested in adults. However, none has been approved for commercial use in the pediatric population [8]. In adult patients, particularly promising results have been shown with evinacumab. This is a first-in-class recombinant human monoclonal antibody that inhibits angiopoietin-like protein 3 (ANGPTL3) developed for treating homozygous familial hypercholesterolemia (HoFH); refractory hypercholesterolemia, both familial and non-familial; and severe hypertriglyceridemia. An intravenous formulation of evinacumab received U.S. Food and Drug Administration (FDA) approval in 2021 as an add-on therapy in HoFH for patients aged 12 years or older [9]. Regarding safety profile and pharmacological toxicity, the drug is usually well tolerated. Thus, in phase 1, a single-dose study, up to 80% reduction in triglyceride levels has been reported in evinacumab-treated adult patients with moderate hypertriglyceridemia [10], prompting the effectiveness of evinacumab as a therapeutic option to lower elevated triglycerides. In June 2021, evinacumab obtained authorization for clinical use by the European Medicines Agency (EMA) under exceptional circumstances because comprehensive data had not been provided on the efficacy and safety of the medicine under normal conditions of use. This approval is reserved for treating rare diseases or because collecting complete information is impossible or unethical [11].

In this paper, we report the case of a 10-year-old boy who presented life-threatening refractory hypertriglyceridemia associated with the concomitant use of ruxolitinib and sirolimus for chronic GVHD. After the failure of the insulin treatment and due to the technical impossibility of performing lipid apheresis, the child underwent evinacumab treatment, obtaining a rapid and dramatic reduction in triglyceride and cholesterol levels. To our knowledge, this is the first-ever description of a pediatric patient younger than 12 years of age with life-threatening hypertriglyceridemia treated in Europe with evinacumab on a compassionate-use basis.

## 2. Case Presentation

A 10-year-old Caucasian boy affected by T-cell acute lymphoblastic leukemia underwent an allogenic HSCT from a matched unrelated donor in May 2020. His post-transplant course was complicated by the chronic hepatic and diffuse sclerotic form of skin GVHD, and he was well controlled on prednisone, tacrolimus, and ruxolitinib. Due to prolonged high-dose corticosteroid treatment associated with ruxolitinib and tacrolimus, the boy developed a metabolic syndrome featured by dyslipidemia, increased insulin resistance, and severe refractory hypertension. During routine periodic assessments, his cholesterol and triglyceride levels were between 300 and 400 mg/dL over the last several months. A liver ultrasound and liver biopsy confirmed the presence of steatosis. No family history of dyslipidemia was reported.

In addition, immediately after HSCT, high-level BK virus (BKV) blood and urine loads were documented. Subsequently, BK viremia stabilized between 2000 and 4000 copies/mL, while viruria fluctuated between 50,000 and 100,000 copies/mL, without any clinical signs.

Two years after transplantation, a routine outpatient follow-up noted an acute kidney injury requiring hospitalization. At admission, the child’s immunosuppressive therapy consisted of low-dose methylprednisolone (15 mg/die, corresponding to 0.5 mg/kg/die), tacrolimus (approximately 0.15 mg/kg/die), and ruxolitinib (5 mg twice daily). Laboratory investigation revealed elevated values of creatinine (1.63 mg/dL, normal range 0.26–0.55 mg/dL), blood urea nitrogen (92 mg/dL, normal range 19–47 mg/dL), and hyperuricemia (8.9 mg/dL, normal range 1.8–4.9 mg/dL). The state of well-known immunodeficiency was confirmed: the number of CD3 was 402 cells/µL, the number of CD4 168 was cells/µL, and there was an absence of CD19. In addition, moderate hypertriglyceridemia was reported (394 mg/dL, with an acceptable value for age <90 mg/dL).

Given the immunosuppression, his renal disease was better investigated. An ultrasound of the urinary tract showed no evidence of obstruction nor hydronephrosis and a normal renal size. PCR for BKV was performed, showing a marked increase in virus load with more than 100 million copies/mL in the urine and more than 25 million copies/mL in the blood. Clinical and laboratory findings were consistent with diagnosing BKV-associated nephropathy, which typically occurs in immunocompromised patients because of viral reactivation, especially when treated with tacrolimus [12]. According to the literature data, a reduction in immunosuppression was performed [13]. The dosages of corticosteroids and ruxolitinib were halved (being reduced to 10 mg/die and 2.5 mg twice daily, respectively), while tacrolimus was replaced with sirolimus at a dose of 0.5 mg/day. Intravenous immunoglobulin (IVIG) was also administered.

As renal function continued to deteriorate with no trend of decreasing BKV load, a single dose of cidofovir (3 mg/kg) was administrated. Cidofovir led to a progressive reduction of BKV blood load, down to 33.000 copies/mL, but caused significant worsening of kidney function, with a creatinine clearance reduction of up to 32 mL/min. Furthermore, the boy developed severe trilineage hematologic toxicity, rapidly leading to profound lymphopenia, thrombocytopenia, and anemia. Particularly, his white blood cell (WBC) count reached a minimum of 810 cells/mm^3^ (normal range 4500–13,500 cells/mm^3^), with absolute lymphocyte count nadir at 310 cells/mm^3^. The lowest platelet value was 12,000 cells/mm^3^ (normal range 150,000–450,000 cells/mm^3^), while hemoglobin levels reached a nadir of 6.9 g/dL (normal range 12.0–14.0 g/dL), causing plasma and red blood cells transfusions to be required. Thus, acute kidney injury was associated with renovascular hypertension (average systolic and diastolic blood pressure were 178/109 mmHg, with the normal range for this age being 102–120 mmHg and 61–80 mmHg, respectively), requiring the withdrawal of corticosteroids and initiation of antihypertensive therapy with propranolol (up to 40 mg three times a day) and amlodipine (10 mg once a day) to reach an adequate blood pressure control.

Two weeks later, our patient was diagnosed with invasive *Scedosporium apiospermum* osteomyelitis, for which he requested surgical debridement with resection of surgically amenable lesions. The clinical picture was rapidly complicated by life-threatening chronic Norovirus enteritis and hepatitis that were beyond systemic HSV-1 and CMV infections with pulmonary and central nervous system involvement. In light of the impossibility of starting antiviral therapy due to kidney failure, the cortisone was stopped in the expectancy of obtaining an immune recovery. The boy remained in combined therapy with sirolimus and ruxolitinib.

An already clinically difficult situation was further complicated by acute pancreatitis. Excluding the various causes, the child’s metabolic profile was studied. Triglyceride levels became extremely high, up to 6000 mg/dL, while cholesterol increased to 2900 mg/dL. In the suspicion that hypertriglyceridemia was due to the combination of sirolimus and ruxolitinib, as previously reported in the literature [5,6], the immunosuppression was discontinued. During toxicity, the patient received 2.5 mg of ruxolitinib twice daily and 0.5 mg of sirolimus once a day. Sirolimus serum levels were tested and were always in the range of normality (ranging from 2.10 to 8.65 ng/mL, with a therapeutic range of 4.00–12.00 ng/mL in the blood). In this setting, adequate blood pressure control was held thanks to combined therapy with propranolol and amlodipine that was previously initiated.

Considering that the severity of dyslipidemia required urgent intervention beyond the simultaneous presence of renal and hepatic insufficiency, we excluded the possibility of using fibrates and statins. Therapeutic apheresis was excluded because of the absence of central line accesses that could guarantee sufficient flow during apheresis. Given the child’s diffuse scleroderma, placing a single central venous catheter of a small gauge was possible. The insulin-dextrose infusion was a unique suitable therapeutic option. The boy was started on a continuous insulin infusion with substantially reduced triglyceride levels, which dropped to 1300 mg/dL in one week. Unfortunately, the patient demonstrated a poor tolerance of insulin treatment due to frequent episodes of hypoglycemia and the impossibility of monitoring glucose levels with the FreeStyle Libre (intermittently scanned continuous glucose monitoring) through his fibrotic skin.

Having obtained the favorable opinion of the Regional Committee for Bioethics for using evinacumab, we asked Ultragenyx Pharmaceutical Inc. (60 Leveroni court, Novato CA, USA). for the medicine supply on a compassionate basis. Within a few days, we received evinacumab (evinacumab-dgnb, EVKEEZA^TM^) from Regeneron Pharmaceuticals, Inc. (777 Old Saw Mill River Rd., Tarrytown, NY, USA). The child started therapy with evinacumab at a dose of 15 mg/kg every four weeks, for a total of 450 mg per dose, with a pre-treatment triglyceride level of 1341 mg/dL. We documented a dramatic reduction of blood triglyceride levels, reaching 154 mg/dL in 24 h. The drug was well-tolerated, and the boy continued the inpatient treatment three months, maintaining triglyceride levels below 350 mg/dL. However, they remained higher than the normal range for his age.

In the meantime, his clinical conditions have progressively improved, and he has been discharged with a complete immunological reconstruction. Eight months later, the patient continues evinacumab outpatient treatment without any adverse event, maintaining markedly reduced triglyceride levels (167 mg/dL at the last follow-up). Figure 1 displays the reduction of triglyceride and total cholesterol values over time.

## 3. Discussion

In our patient, we assumed that markedly high triglyceride levels were derived from the combined therapy with sirolimus and ruxolitinib. Dyslipidemia, particularly hypertriglyceridemia, is a major adverse effect of sirolimus described in the literature [14]. Thus, a mild increment in triglyceride levels without any significant clinical outcome has been described in ruxolitinib use [15]. The association between these two immunosuppressors increases the risk of hypertriglyceridemia and metabolic syndrome exponentially, as previously reported in the literature both in adult and in pediatric patients [5,6]. The other adverse events described in this kind of combined therapy refer to those described in single agents’ therapy, such as hemorrhagic cystitis, mucositis, and pulmonary fungal infections [16].

The 2018 American College of Cardiology/American Heart Association cholesterol-levels guidelines define “severe hypertriglyceridemia” as triglyceride levels being ≥500 mg/dL [17]. The Endocrine Society further specifies, defining values of 2000 mg/dL as very severe hypertriglyceridemia [18].

The guidelines for managing severe hypertriglyceridemia in adults recommend using fibrates (most commonly fenofibrate) as first-line treatment if creatinine clearance is >30 mL/min and liver transaminases values are <2.5 × the upper limit of the normal. Furthermore, omega-3 fatty acids are recommended, either alone or with statins [17]. However, hypolipidemic drugs fail to reduce triglyceride-rich lipoproteins in patients with hypertriglyceridemia-induced pancreatitis or severe hyper-viscosity symptoms. In these cases, an insulin-dextrose infusion or therapeutic apheresis should be recommended [19].

Managing severe hypertriglyceridemia in children is challenging because of the lack of evidence-based guidelines. Furthermore, no FDA/EMA-approved triglycerides-lowering drugs have been approved for commercial use in pediatrics. Some novel lipid-lowering medications acting on the lipoprotein lipase (LPL) complex have recently emerged in the adult population. More specifically, LPL is an extracellular enzyme located on the vascular endothelial surface, mainly represented in adipose tissue, cardiac and skeletal muscles, and macrophages. LPL is absent in the liver, as the liver tissue has specific hepatic lipase. LPL plays a major role in lipid metabolism, being responsible for the hydrolysis of the triglycerides and phospholipids present in circulating plasma lipoproteins, such as chylomicrons and very-low-density lipoproteins (VLDL), producing chylomicron remnants and intermediate-density lipoproteins (IDLs), respectively [20].

In the last few years, the metabolic role of some molecules belonging to the family of angiopoietin-like proteins (ANGPTLs) has been emerging. Particularly, angiopoietin-like protein 3 (ANGPTL3), a glycoprotein secreted by the liver, was found to play a key role in the metabolism of plasmatic lipoproteins. ANGPTL3 is a physiological inhibitor of LPL and endothelial lipase (EL), increasing plasmatic levels of triglycerides, LDL cholesterol, and HDL cholesterol (combined hyperlipidemia) [21]. Thus, Dewey et al. documented that triglyceride levels were notably reduced in carriers of loss-of-function mutations in the ANGPTL3 encoding gene, suggesting that it could be a possible therapeutic target for lowering elevated triglycerides [22]. These observations generated the hypothesis that ANGPTL3 inhibitors could represent a novel therapeutic strategy in treating hypertriglyceridemia and combined hyperlipidemia, mimicking loss-of-function mutations.

Evinacumab, a fully human monoclonal antibody that acts as an ANGPTL3-inhibitor, received FDA approval in 2021 as an add-on therapy in homozygous familial hypercholesterolemia (HoFH) for patients aged 12 years or older [9]. Currently, a three-part, single-arm, open-label clinical trial is evaluating the efficacy and safety of evinacumab in children with HoFH between the ages of 5 and 11 years (NCT04233918) [23]. Despite the data collected from two phase 1, single-dose studies, which have demonstrated a dose-dependent reduction of up to 80% in triglyceride levels in evinacumab-treated adults, ANGPTL3-inhibitors have not received FDA approval as a triglyceride-lowering drug [24]. In HoFH, the FDA-recommended dosage of evinacumab is 15 mg/kg, administered intravenously for one hour once every four weeks [24]. The mechanism of action of evinacumab is shown in Figure 2.

Treatment is generally well-tolerated. In an eight-month follow-up, our patient did not present any of the most common side effects previously described, such as hypersensitivity reactions, infusion site reactions, flu-like symptoms, nasal congestion, abdominal pain, diarrhea, and elevation of transaminases [25,26].

## 4. Conclusions

To our knowledge, we describe the first pediatric patient younger than 12 years of age to receive evinacumab to treat severe hypertriglyceridemia on a compassionate-use basis. In our experience, therapy with evinacumab was effective, safe, and well-tolerated. It allowed us to reduce triglyceride levels quickly, avoiding more invasive and unsafe therapeutic options, such as prolonged continuous insulin infusion or therapeutic plasma exchange. There are no current FDA/EMA-approved triglyceride-lowering drugs in pediatric patients. We hope that our case report can represent a breakthrough in treating life-threatening hypertriglyceridemia in children affected by hematological malignancies, with numerous severe complications due to treatments.

Furthermore, we would like to stress the importance of treating and preventing the underlying causes leading to secondary hypertriglyceridemia. In a transplantation setting, in agreement with the reports of Watson and Bauters [5,6], we emphasize the need to monitor severe side effects in immunosuppressive drugs combination, especially regarding the patient’s metabolic status.

## Figures and Tables

**Figure 1 pharmaceuticals-16-01069-f001:**
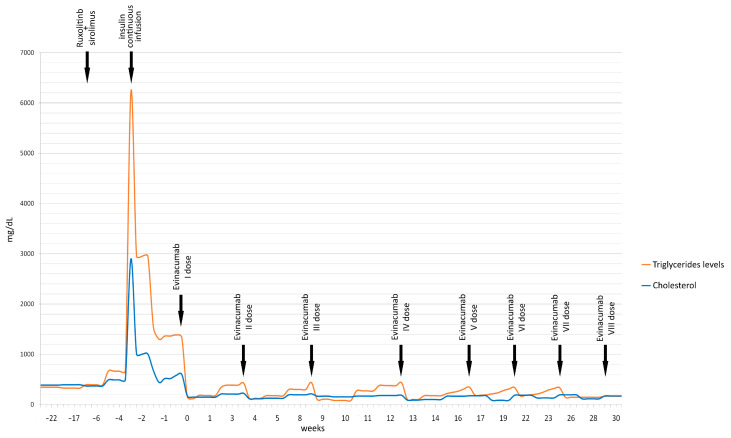
Timeline of patient’s triglyceride and cholesterol levels (mg/dL) before and after the start of evinacumab treatment.

**Figure 2 pharmaceuticals-16-01069-f002:**
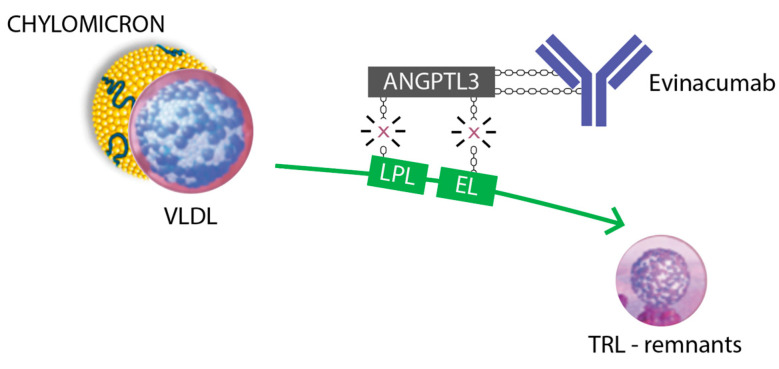
ANGPTL3 is reported to affect plasma triglyceride-rich lipoproteins (VLDL and chylomicrons) by inhibiting LPL, especially in the fed state, and HDL metabolism through inhibiting EL-mediated phospholipid hydrolysis. Abbreviations: VLDL, very-low-density lipoprotein; LPL, lipoprotein lipase; EL, endothelial lipase; TRL, triglyceride-rich lipoproteins.

## Data Availability

The original contributions presented in this case report are included in the article. Further inquiries can be directed to the corresponding author.

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
