# Peer review of "Rapid Resolution of Life-Threatening Hypertriglyceridemia after Evinacumab Administration in a Pediatric HSCT Recipient: A Case Report"

_pharmaceuticals, 2023, doi:10.3390/ph16081069_

Round 1

Reviewer 1 Report

What is the standard first-line treatment for chronic graft-versus-host disease (GVHD)?

Which drugs have shown promising efficacy as second-line systemic treatments for corticosteroid refractory chronic GVHD?

What are the adverse effects commonly associated with ruxolitinib and sirolimus combination therapy?

What is the mechanism of action of evinacumab and what is its approved indication?

What are the recommended treatment options for managing severe hypertriglyceridemia in adults and children, respectively, according to the guidelines mentioned in the paper?

Author Response

  • What is the standard first-line treatment for chronic graft-versus-host disease (GVHD)?
    • In order to better understand this point, we rewrote this part of the introductive section as such: “Historically, systemic corticosteroids have been the standard first-line treatment for chronic GVHD, thanks to their lymphopenic and anti-inflammatory properties; particularly, a starting dose of 1 mg/kg/die of prednisone (or an equivalent dose of methylprednisolone) has typically been used in this setting. Systemic glucocorticoid therapy could me given alone or in association with other immunosuppressive agents, such as calcineurin inhibitors (e.g. tacrolimus), which typically prevent IL-2 production by T-cells.”
  • Which drugs have shown promising efficacy as second-line systemic treatments for corticosteroid refractory chronic GVHD?
    • In order to better understand this point, we rewrote this part of the introductive section as such: “The choice of a specific drug as a second-line treatment in chronic GVHD generally depends on patients’ comorbidities, specific organ involvement in the graft, and clinicians’ experience. Particularly, mTOR inhibitors (e.g. sirolimus), Janus kinase (JAK) inhibitors (e.g. ruxolitinib and baricitinib), Bruton’s tyrosine kinase inhibitors (e.g. ibrutinib), proteasome inhibitors (e.g. bortezomib), and anti-CD20 monoclonal antibodies (e.g. rituximab) have shown promising efficacy with an adequate safety profile”
  • What are the adverse effects commonly associated with ruxolitinib and sirolimus combination therapy?
    • To our knowledge, the association between these two immunosuppressors increases the risk of hypertriglyceridemia and metabolic syndrome exponentially, as previously reported in the literature both in adult and in pediatric patients [Watson AP, Brunstein CG, Holtan SG. Life-Threatening Hypertriglyceridemia in a Patient on Ruxolitinib and Sirolimus for Chronic Graft-versus-Host Disease. Case Rep Transplant. 2018;2018:4539757, Bauters T, Bordon V, Laureys G, Dhooge C. Combined use of ruxolitinib and sirolimus: increased monitoring of triglycerides required. Bone Marrow Transplant. 2019;54(8):1372-1373]. The other adverse events described in this kind of combined therapy refer to the ones described in single agents’ therapy, such as hemorrhagic cystitis, mucositis, and pulmonary fungal infections [Wang, Sanbin & Huang, Xiaoli & Liu, Lin & Liang, Yang. (2018). Outcome of Ruxolitinib and Sirolimus in Preventing aGVHD Post to HLA Matched Hemotopietic Stem Cell Transplantation. Blood. 132. 5727-5727]. We add these details in the main text, starting on line 190.
  • What is the mechanism of action of evinacumab and what is its approved indication?
    • Evinacumab is a fully human monoclonal antibody that acts as an ANGPTL3-inhibitor, received FDA approval in 2021 as an add-on therapy in homozygous familial hypercholesterolemia (HoFH) for patients aged 12 years or older [Markham A. Evinacumab: First Approval. Drugs. 2021;81(9):1101-1105]. Currently, a three-part, single-arm, open-label clinical trial is evaluating the efficacy and safety of evinacumab in children with HoFH between the ages of 5 and 11 years (NCT04233918) [ClinicalTrials.gov Identifier: NCT04233918. A three-part, single-arm, open-label study to evaluate the efficacy, safety, and pharmacokinetics of evinacumab in pediatric patients with homozygous familial hypercholesterolemia - Available from: https://clinicaltrials.gov/ct2/show/NCT04233918, accessed March 13, 2022]. In order to better understand the pathophysiology, we rewrote the previous part as such: “Some novel lipid-lowering medications acting on the lipoprotein lipase (LPL) complex have recently emerged in the adult population. More specifically, LPL is an extracellular enzyme located on the vascular endothelial surface, mainly represented in adipose tissue, cardiac and skeletal muscles, and macrophages. LPL is not present in the liver, as the liver tissue has specific hepatic lipase. LPL plays a major role in lipid metabolism, being responsible for the hydrolysis of the triglycerides and phospholipids present in circulating plasma lipoproteins, such as chylomicrons and very low-density lipoproteins (VLDL), producing chylomicron remnants and intermediate-density lipoproteins (IDLs), respectively. In the last few years, the metabolic role of some molecules belonging to the family of the Angiopoietin-Like Proteins (ANGPTLs) is emerging. Particularly, Angiopoietin-Like Protein 3 (ANGPTL3), a glycoprotein secreted by the liver, was found to play a key role in the metabolism of plasmatic lipoproteins. ANGPTL3 is a physiological inhibitor of LPL and endothelial lipase (EL), increasing plasmatic levels of triglycerides, LDL cholesterol, and HDL cholesterol (combined hyperlipidemia). Thus, Dewey et al. documented that triglyceride levels were notably reduced in carriers of loss-of-function mutations in the ANGPTL3 encoding gene, suggesting it could be a possible therapeutic target for lowering elevated triglycerides. These observations contributed to generate the hypothesis that ANGPTL3 inhibitors could represent a novel therapeutic strategy in treating hypertriglyceridemia and combined hyperlipidemia, in an attempt to imitate loss-of-function mutations.” After this paragraph, we specified Evinacumab mechanism of action, whose mechanism of action is reported in Figure 2 in the main text.
  • What are the recommended treatment options for managing severe hypertriglyceridemia in adults and children, respectively, according to the guidelines mentioned in the paper?
    • As we report in the main text, the guidelines for managing severe hypertriglyceridemia in adults recommend using fibrates (most commonly fenofibrate) as the first-line treatment if creatinine clearance is > 30 ml/min and liver transaminases values are < 2.5 x the upper limit of the normal. Furthermore, omega-3 fatty acids are recommended, alone or with statins. However, hypolipidemic drugs fail to reduce triglyceride-rich lipoproteins in patients with hypertriglyceridemia-induced pancreatitis or severe hyper-viscosity symptoms. In these cases, insulin-dextrose infusion or therapeutic apheresis should be recommended. Managing severe hypertriglyceridemia in children is challenging because of the lack of evidence-based guidelines. Furthermore, no FDA/EMA-approved triglycerides-lowering drugs have been approved for commercial use in pediatrics.

Reviewer 2 Report

Unique and interesting case report showing usefullness of evinacumab in a young boy with severe hypertrigluceridemia following HCT and therapy for chronic GVHD with tacrolimus and ruxolitinib.

Some suggestions.What were the ranges of blood pressure during hypertriglyceredemia?What were doses of prednisolone,tacrolimus and ruxolitinib?What were tacrolimus serum levels during hypertriglyceremia?What were doses of Sirolomus?Give range and median doses.What were WBC,platletsand HB levels,ranges during toxicity?

Minors.On line 75 replace e with and.

Change HSCT to HCT,hematopoietic cell transplantation.

Author Response

We thank the reviewer for the kind words. As for the relevant questions:

  • What were the ranges of blood pressure during hypertriglyceridemia?
    • We report blood pressure values both during acute kidney injury and hypertriglyceridemia. Particularly, we rewrote these two parts as such: 1. “Thus, acute kidney injury was associated with renovascular hypertension (average PAS/PAD 178/109 mmHg, with normal range for age 102 – 120 mmHg and 61 – 80 mmHg, respectively), requiring the withdrawal of corticosteroids and initiation of antihypertensive therapy with propranolol (up to 40 mg three times a day) and amlodipine (10 mg once a day), reaching an adequate blood pressure control”; 2. “In this setting (referring to hypotriglyceridemia, n.d.r.), an adequate blood pressure control (PAS/PAD 120/75 mmHg) was held thanks to combined therapy with propranolol and amlodipine.”
  • What were doses of prednisolone, tacrolimus and ruxolitinib?
    • At the admission, the child’s immunosuppressive therapy consisted of low-dose methylprednisolone (15 mg/die, corresponding at 0.5 mg/kg/die), tacrolimus (approximately 0.15 mg/kg/die), and ruxolitinib (5 mg twice per day). We added drug doses in the main text (starting from line 100).
  • What were tacrolimus serum levels during hypertriglyceridemia?
    • During toxicity, the patient was receiving ruxolitinib 2.5 mg twice daily, and sirolimus 0.5 mg once a day, since tacrolimus was replaced with sirolimus during BKV infection. For this reason, during hypertriglyceridemia, tacrolimus serum levels were 0.
  • What were doses of Sirolomus? Give range and median doses.
    • During BKV infection, tacrolimus was replaced with sirolimus at a dose of 0.5 mg/day (we reported it on line 117). On line 147, we rewrote the paragraph reporting sirolimus serum levels: “During toxicity, the patient received ruxolitinib 2.5 mg twice daily, and sirolimus 0.5 mg once daily. Sirolimus serum levels were texted, being always in the range of normality (ranging from 2.10 to 8.65 ng/mL, with a therapeutic range of 4.00 – 12.00 ng/mL in the blood).”
  • What were WBC, platelets and Hb levels, ranges during toxicity?
    • About toxicity, we rewrote this part as such: “Furthermore, the boy developed severe trilineage hematologic toxicity, rapidly leading to profound lymphopenia, thrombocytopenia, and anemia. Particularly, white blood cells (WBCs) count reached a minimum of 810 cells/mmc (normal range 4500 – 13500 cells/mmc), with absolute lymphocyte count nadir at 310 cells/mmc. The lowest platelets value was of 12.000/mmc (normal range 150000 – 450000/mmc), while hemoglobin levels reached a nadir of 6.9 g/dL (normal range 12.0 – 14.0 g/dL), requiring plasma and red blood cells transfusions.”
    • On line 75 replace e with and.
      • Changes have been made accordingly.
    • Change HSCT to HCT, hematopoietic cell transplantation.
  • With all due respect to the reviewer's comment, we have decided not to replace the term HSCT with HCT. The term HSCT (hematopoietic stem cell transplantation) is universally accepted by the transplant community worldwide and is a term created by the EBMT (European Bone Marrow Transplant Group). Hematopoietic stem cell transplantation is the exact term used to describe the transplant procedure which our patient underwent.